# Phytogenic Antioxidants Prolong n-3 Fatty Acid-Enriched Eggs’ Shelf Life by Activating the Nrf-2 Pathway through Phosphorylation of MAPK

**DOI:** 10.3390/foods11203158

**Published:** 2022-10-11

**Authors:** Muhammad Suhaib Shahid, Shengyu Zhou, Wei Nie, Liang Wang, Huiyuan Lv, Jianmin Yuan

**Affiliations:** 1State Key Laboratory of Animal Nutrition, College of Animal Science and Technology, China Agricultural University, Beijing 100193, China; 2Beijing General Station of Animal Husbandry, Beijing 100101, China; 3Beijing Center of Biology Co., Ltd., Beijing 102206, China

**Keywords:** α-linolenic acid (ALA), docosahexaenoic acid (DHA), lipid oxidation, gene expression, magnum morphology

## Abstract

Helpful for human health, omega-3 (n-3)-enriched eggs are preferred by consumers. However, antioxidants should be added to the hen’s diet to prevent n-3 fatty acid oxidation due to their unsaturated bonds. A study was designed to investigate the effects of different antioxidants on performance, egg quality, fatty acid profile, oxidation parameters, gene expression, and magnum morphology. A total of 450 hens were divided into five dietary groups. Wheat–flaxseed was used for the basic diet (control) and supplemented with vitamin E (VE), chlorogenic acid (CA), polyphenol (PF), and lutein (L). The experiment lasted for 10 weeks. The eggs were collected on the 5th week and were analyzed for quality, oxidative stability, and fatty acid (FA) content, being stored for 0 d, 7 d, 14 d, 21 d, 28 d, 35 d, and 42 d. The results showed that supplemental VE, PF, CA, and L improved the egg weight and hen day egg production compared to the control group (*p* < 0.05). The VE, PF, and L groups significantly (*p* < 0.05) reduced the malondialdehyde (MDA) and maintained the superoxide dismutase (SOD), glutathione peroxidase (GSH-Px), and total antioxidant capacity (T-AOC) in the egg yolk. The albumen height and Haugh unit were maintained in the egg yolk till 35 days of storage by the VE, PF, and L groups, while the CA group reduced the albumen quality after 21 d storage. The VE, PF, CA, and lutein maintained the content of alpha-linolenic acid (ALA), during the whole storage period. The total n-3 FA and docosahexaenoic acid (DHA) were retained in the egg yolk till 35 and 28 days of storage, respectively, and slightly decreased after 35 and 28 days in the L groups. The total n-6 (Tn-6) FA was maintained in the yolk till 28 days of storage in the CA and PF groups, respectively. The VE, PF, and L groups upregulated the expression of Nrf-2, P38MAPK, HO-1, SOD-1, and GSH-Px as compared to the CA and control groups. The VE, PF, and L groups significantly increased the magnum primary folds and epithelium height as compared to CA and the control. Thus, it was concluded that the use of PF and L is better at preventing egg quality deterioration and lipid oxidation, maintaining more than 300 mg/egg n-3 FA during storage, by activating the Nrf-2 pathway through the phosphorylation of P38MAPK, and enhancing the phase-2 antioxidant defense enzymes, namely, SOD, GSH-Px, and HO-1.

## 1. Introduction

Eggs enriched with n-3 polyunsaturated fatty acid (PUFA) can be helpful for human health. Different sources of n-3, mainly flaxseed, are used to produce n-3 eggs. However, n-3-enriched feeds, such as flaxseed, are easily oxidized, which harms the health and performance of poultry [1]. The n-3-enriched eggs are prone to lipid oxidation due to the presence of unsaturated bonds [2]. During longer storage, reactive oxygen species (ROS) are produced in the eggs and ameliorate albumen quality [3]. The oxidative stress also causes morphological changes in the albumen, which affects the egg quality [4]. Moreover, toxic compounds are produced from the oxidation of n-3 FA-enriched products during transport, storage, and consumption [5]. So, antioxidants, usually 100 IU/kg vitamin E, are added to the hens’ diet to prevent oxidation in n-3-enriched eggs and improve the shelf life of n-3 PUFA-enriched eggs for consumer acceptance [6]. Previous studies used vitamin E and selenium to maintain egg quality during storage [7,8]. However, the research found supplemental vitamin E retains the shelf life of n-3 FA in eggs only for 3 weeks of storage [9]. In addition, a high level of vitamin E will compete with the fat-soluble vitamins, such as vitamin D absorption and utilization [10]. So, a new, high-activity antioxidant needs to be used to improve the shelf life of n-3-enriched eggs.

Previous studies reported that natural antioxidants can help reduce lipid oxidation in the egg powder during storage [11]. Tea polyphenols are extracted from green tea, which can improve the laying performance of hens and egg quality [12]. Lutein is a carotenoid in nature and has an outstanding antioxidant ability [13]. The main active ingredient of some natural herbs, chlorogenic acid, has found potent antioxidant activity [14], which could increase the serum antioxidant profile and mRNA expression of antioxidant genes, namely, SOD, GSH-Px, and catalase, of hens [15]. However, which antioxidant can be used to replace vitamin E in a hen’s diet is not known.

Antioxidant compounds exert their antioxidant activity by scavenging free radicals, metal chelation, inhibiting cellular proliferation, modulation of enzymatic activity, and signal transduction pathways [16]. In some studies, the addition of antioxidants can enhance the endogenous antioxidant defenses in the target organs, such as the oviduct of laying hens [17]. Research showed that antioxidants effectively reduce oxidative stress via the activation of the Nrf2/HO-1 pathway [18]. The activation of the Nrf2/HO-1 pathway enhanced the activity of its effectors, such as mitogen-activated protein kinases (MAPKs) [19]. MAPKs regulate the expression of HO-1 through activation of Nrf2 translocation [20], and HO-1 can alleviate oxidative stress through scavenging activity [21]. However, little is known about the antioxidant effects and the mechanism of polyphenol, lutein, and chlorogenic use in the wheat–flaxseed diet. Therefore, this study was designed to evaluate the potential role of supplementing flaxseed with natural antioxidants in the performance, egg quality, fatty acid and antioxidant profile of eggs during storage, and activating the Nrf-2 pathway and its related genes expression in the liver and magnum of laying hens.

## 2. Materials and Methods

### 2.1. Animal Ethics

All procedures and practices of the experiment were approved by the Animal welfare and Ethical Committee for Laboratory Animals, China Agricultural University, and conducted per the guidelines for experimental animals (No. AW04110202-2-1).

### 2.2. Antioxidants Description

The chlorogenic acid was extracted from *Lonicera japonica* (2.4 g/1000 g of chlorogenic acid). The polyphenol was purchased from Methodo Chemicals Italy (Novellara, Italy); it was a mix of vegetable polymers (polyphenol) obtained from “Pine-wood”, purified, hydrolyzed, with a low molecular weight from 500 to 5000 Dalton. The lutein (which contains 20.0 g of xanthophyll activity per kg) was bought from Kemin Technologies Co, Ltd., Zhuhai, China. The vitamin E (1 IU/mg) was purchased from Zhejiang Pharmaceutical Co., Ltd. (Zhejiang, China).

### 2.3. Birds’ Husbandry

A graphical representation of the experimental scheme is given in Figure 1. A total of 450, 36-week-old, white Nongda-3 hens were divided into five dietary treatments. Hens were kept in cages. Every treatment group had six replicates, each having 15 hens. Five conjoint cages with 3 hens in each cage were regarded as the same replicate. The basal diet of the wheat–flaxseed diet was formulated to fulfill the nutrient requirements of laying hens according to the recommendations of the NRC [22] (Table 1). The treatments were supplemented with vitamin E (VE 0.02%), chlorogenic acid (CA 0.24%), polyphenol (PF 0.05%), and lutein (L 0.03%), balanced by zeolite powder. The level of antioxidants was selected according to the company’s recommended levels. The feed and water were provided ad libitum. The birds were given 16 h of light. The experiment lasted for 10 weeks.

### 2.4. Performance Parameters

Egg number and egg weight were recorded daily, and feed intake (FI) was recorded at the end of every two weeks. The feed conversion ratio (FCR) was calculated based on egg mass.

### 2.5. Sampling for Egg Storage—Albumen Quality, Lipid Oxidation, and Fatty Acid Profile

On the 5th week of the experiment, 21 eggs from each replicate (3 eggs/replicate × 7 storage points = 21 eggs) were collected and stored at room temperature for 0, 7, 14, 21, 28, 35, and 42 days. The temperature of the room recorded from 7 to 42 days of storage was 4–6 °C. From each storage period, three eggs from each replicate were collected to measure the albumen quality. The albumen height and Haugh unit were analyzed by using Digital EGG TESTER (DET-6000). Egg yolks were then separated, rolled on filter paper, pooled (3 egg yolks), and freeze-dried at −80 °C. The pooled egg yolks were divided into two parts and stored at −80 °C; one was analyzed for lipid oxidation (MDA, SOD, GSH-Px, T-AOC) and the other was used for the fatty acid profile of the egg yolk during storage.

#### 2.5.1. Oxidative and Antioxidant Activities

Lipid oxidation markers, such as thiobarbituric acid reactive substances, are expressed as malondialdehyde (MDA) equivalents. The MDA and the activities of antioxidant glutathione peroxidase (GSH–Px), catalase (CAT), and superoxide dismutase (SOD) in egg yolks were determined using the commercial assay kits purchased from Nanjing Jiancheng Institute of Bioengineering (Jiangsu, China), following the standard procedures described by the manufacturer. [22]. The water used in the chemical analysis was ultra-purified.

#### 2.5.2. Fatty Acid Analysis

Fatty acid methyl esters (FAME) of stored and fresh egg yolks were prepared using direct FAME synthesis, as described by Christe [23]. Briefly, 0.1 g of egg yolk was weighed and added to a 50 mL tube; subsequently, 4 mL of chloro-acetyle methanol (1:10), 1 mL of hexane, and 0.5 mL of an ethanol standard solution were added, and the lid closed tightly. The tubes were placed in a water bath for 3 h at 75 °C. Tubes were cooled, 4 mL of potassium carbonate solution was added, vortexed (1 min), and centrifuged for 5 min at 900 RPM. Supernatant solution (1 mL) was taken in small GC tubes. The fatty acid composition of the FAME was determined by a capillary gas chromatograph (GC) on an SP-2560, 100 m × 0.25 m × 0.20 m capillary column installed on a Hewlett-Packard 5890 gas chromatograph equipped with an HP 3396 injector and HP 7673 controller, a flame ionization detector, and split injection. The initial oven temperature was 140 °C, held for 5 min, and then increased to 240 °C at a rate of 4 °C/min. Helium was used as the carrier gas at a flow rate of 0.5 mL/min. Both the injector and the detector were set at 260 °C. Fatty acids were identified by comparing their retention times with the standard Supelco 37 Component FAME Mix C_4_-C_24_ (Sigma-Aldrich, Hamburg, Germany). The undecanoic fatty acid (C11:0) was used as an internal standard to allow the conversion of the relative percentage (%FA/total FA) of each fatty acid in absolute value, as mg/egg.

### 2.6. Sampling and Preparation for Liver and Magnum Analysis

By the end of the feeding trial, on the 11th week, one hen from each replicate was randomly chosen, and sacrificed by cutting the left jugular vein. The liver and the whole length of the oviduct were removed, and approximately 2 cm of the tissue samples from the midpoint of the magnum was obtained and flushed with cold phosphate-buffered saline (PBS). The parts of the sampled magnum and liver were frozen using liquid nitrogen and stored at −80 °C for further analysis. Another was used for morphology measurements.

### 2.7. RNA Extraction and Reverse Transcription

Total RNA of the liver and magnum sample was extracted using Trizol Reagent (Invitrogen Biotechnology Inc., Carlsbad, CA, USA) according to the manufacturer’s protocol. The primers for genes are shown in Table 2. The expression of *Nrf1*, *SOD1*, *GSH-Px CAT*, *HO-1*, and *P38MAPK* were measured by real-time PCR for measuring the expression of genes in the liver and jejunum, carried out using SYBR Premix Ex Taq (TliRNaseH Plus) (Takara Biotechnology Inc., Osaka, Japan) in an ABI 7500 Real-time PCR Systems (Applied Biosystems, Foster City, CA, USA).

### 2.8. Magnum Morphology

The samples were lightly flushed numerous times with physiological saline (0.1% NaCl) to eliminate their contents and placed in 10% formalin in 0.1 M phosphate buffer (pH = 7.0) for fixation (48 h). The samples were treated for 24 h in a tissue processor with anhydrous ethanol as the de-hydrant, and then the samples were implanted in paraffin. Three 4-μm slices were made from the tissue and stained with hematoxylin and eosin. The magnifications were viewed at 100× and 400× using an analyzer (Nikon DS-U3, Tokyo, Japan) coupled with a digital camera (BX51; Olympus Corp., Tokyo, Japan). The longitudinal section morphometry contained the epithelial cell height of the simple columnar epithelium. The height of a primary fold was measured by drawing a vertical line from the base to the luminal end [4]. Epithelial cell height was determined by measuring the height of 15 cells in 5 different primary folds. Each treatment had 6 replicates with 1 bird each, and 3 samples were examined for each bird, with 2 images taken of each sample.

### 2.9. Statistical Analysis

Data were analyzed as a completely randomized design by the SPSS 20.0 [24]. One-way ANOVA was used to study the effect of different dietary antioxidant groups on the performance, albumen quality, egg oxidation parameters, fatty acid profile, gene expression, and magnum morphology. Post-hoc multi-comparisons were applied using Duncan’s test to compare the means of the dietary treatment groups. Moreover, to compare the effect of feeding different dietary antioxidants from 2–10 weeks on the performance of hens and to study the effect of storage time (0–42 d) on the albumen quality, oxidation status, and fatty acid profile of eggs, one-way ANOVA was applied followed by a post-hoc Duncan’s test. The linear and quadratic polynomial contrasts were also used to compare the means of each treatment during eggs storage analysis (0–42 d) or hen’s performance (2–10 weeks) to evaluate whether the differences found followed an increasing or decreasing trend. Significance was set at *p* < 0.05.

## 3. Results

### 3.1. Performance

The effects of various antioxidant sources on the performance of laying hens were represented in Table 3. Supplemental VE, CA, PF, and lutein significantly increased the egg weight (*p* < 0.05), the hens’ egg production, compared with the control, and no significant difference between VE, CA, PE, and lutein was observed. There was no significant difference (*p* > 0.05) in the FCR during the 10-week experimental period. The feed intake was only significant during the 4th week, where FI was reduced in the PF and L groups. The egg production was linearly increased in the CA and L groups after the 2nd week of the experiment. The feed intake was linearly increased after 4 weeks of the experiment in the L group. (Table 3).

### 3.2. Egg Quality

Supplemental antioxidants significantly (*p* < 0.005) improved the albumen height and Haugh unit of the egg compared to the control during storage (Table 4). The albumen height and Haugh unit of the egg in the control linearly decreased with the storage time; however, supplemental VE, PF, and lutein could prevent albumen height from deterioration with the storage time, and no difference was observed among VE, PF, and lutein. Supplemental CA decreased the albumen height from 28 days of storage (*p* < 0.05).

The effect of antioxidants on the Haugh unit was different from the albumen height. Though the Haugh unit of albumen was linearly decreased with stored time, supplemental antioxidants could delay the deterioration of the Haugh unit of eggs. However, there was a difference in the Haugh unit among different antioxidants. Supplemental VE could prevent Haugh from deterioration until stored for 28 days, and CA and PF prevent it until 21 days; however, there were no significant differences for the Haugh unit of an egg after being stored for 28 days, 35 days, and 42 days among VE, PF, and lutein, except CA stored for 28 days.

### 3.3. Oxidation in Egg Yolk during Storage

The various types of antioxidants significantly (*p* < 0.005) affected the oxidation parameters in the egg yolk during storage (Table 5).

Supplemental VE, CA, PE, and L could significantly decrease the content of MDA and increase the content of SOD and T-AOC, and the activity of GSH-Px of the egg yolk compared to the control at 0 days of storage (*p* < 0.05). The MDA content of egg yolk was linearly increased with the stored days, and SOD, T-AOC, and GSH-Px decreased linearly with the stored days. Supplemental VE, CA, PE, and lutein could prevent the oxidation parameters of the egg yolk from deterioration.

Supplemental CA only increased the MDA content at 42 days of storage and decreased SOD at 42 days of storage, and T-AOC and GSH-Px at 35 days of storage.

Supplemental PF decreased SOD in eggs stored for 42 days and decreased GSH-Px stored for 35 days. No difference was observed for MDA and T-AOC.

Supplemental lutein only increased the MDA content at 42 days of storage, decreased T-AOC in eggs stored for 42 days, and increased GSH-Px stored for 35 days. For SOD, no difference was observed.

There were no differences in the content of SOD, T-AOC, and GSH-Px for egg yolks at all stored days among supplemental VE, CA, PE, and lutein. However, supplemental PF could significantly increase the MDA content, more than VE, CA, and lutein (*p* < 0.05).

### 3.4. Fatty Acid Profile of Egg Yolk during Storage

The antioxidant sources significantly (*p* < 0.05) affected the fatty acid profile of egg yolk during storage (Table 6).

Supplemental VE, CA, PE, and lutein could significantly increase the content of DHA, ALA, Tn-3, and Tn-6 while decreasing the FA n-6/n-3 ratio of egg yolk compared to the control eggs stored at 0 days (*p* < 0.05). The Tn-6 of the egg yolk was significantly reduced in the supplemental VE, CA, PE, and lutein yolk compared to the control eggs stored at 0 days (*p* < 0.05). The FA n-6/n-3 ratio of the egg yolk were significantly reduced in the supplemental VE (1.26), CA (1.27), PE (1.27), and lutein (1.27) yolk compared to the control eggs (1.31) stored at 0 days (*p* < 0.05). The content of DHA, ALA, Tn-6, and T n-3 of egg yolk was linearly decreased with stored days in the control, while the FA n-6/n-3 ratio was increased from 1.31 to 1.35 in the control group from Day 0 to 42. Supplemental VE, CA, PE, and lutein could prevent the content of DHA, ALA, T-n3, and Tn-6 of egg yolk from reducing.

Supplemental VE, CA, and PF did not decrease the content of DHA and Tn-3 content at stored days; however, supplemental lutein significantly decreased the DHA content stored at 35 days and decreased the Tn-3 content stored at 42 days. The VE, CA, PF, and lutein groups did not decrease the ALA content during 42 days of storage.

Supplemental VE and L did not decrease the content of Tn-6 at stored days; however, supplemental CA and PF significantly decreased the Tn-6 contents stored at 28 days.

There were no differences in the content of DHA and Tn-3 of egg yolk at all stored days among supplemental VE, CA, and PE. However, supplemental lutein could significantly decrease the DHA and n-3 content after being stored at 35 or 42 days compared to VE, CA, and PF (*p* < 0.05). There was no difference in the content of ALA of egg yolk at all stored days among supplemental VE, CA, PF, and lutein.

### 3.5. Gene Expression in the Magnum and Liver

Dietary antioxidants significantly (*p* < 0.05) affected the Nrf-2 pathway and its related gene expression in the liver and magnum (Table 7). Supplemental VE, CA, PF, and lutein could significantly increase HO-1, SOD-1, GSH-Px, Nrf-2, and P38MAPK gene mRNA expression, both in the magnum and liver, and the CAT mRNA gene expression, only in magnum, compared to the control. However, there was no difference between VE, PF, and lutein.

### 3.6. Magnum Morphology

The folded height and epithelium height in the magnum of hens were significantly (*p* < 0.05) altered by the different sources of antioxidants (Table 8). Supplemental VE, CA, PF, and lutein could significantly increase fold height, EP height, and cilia height compared to the control. There was no difference among VE, PF, and lutein; however, all of them were significantly higher than CA (*p* < 0.05).

Hematoxylin-and-eosin-stained, 100× and 400× images of the magnum are shown in Figure 2. The control group has numerous vacuoles in the degenerating gland cells as well as the degenerating luminal epithelial cells in the control. In the columnar epithelium, there are ciliated cells in the VE, CA, PF, and L groups. In disparity, no cilia were found on the columnar epithelium cells. The gland cells contained round nuclei and an eosinophilic granular cytoplasm can be seen in the VE, CA, PF, and L groups as compared to the control group.

## 4. Discussion

Flaxseed has the highest content of ALA among plant sources of n-3 PUFA. Therefore, it is used in laying hens’ diets for producing n-3-enriched eggs for fulfilling consumers’ demands [25]. However, n-3-enriched eggs are prone to lipid oxidation due to the presence of unsaturated bonds. So, antioxidants are added to the hens’ diet to prevent oxidation in n-3-enriched eggs.

Natural antioxidants have biologically dynamic components that are accomplished by providing welfare to the poultry health and performance. The addition of various antioxidants showed a variation in performance results. In the present study, as compared to the FS diet, the FS with antioxidants such as vitamin E, chlorogenic acid, polyphenol, and lutein enhanced egg production and egg weight. A previous study reported no significant change in egg production when grape seed polyphenols were added to the diet of hens [26]. However, tea-polyphenol addition enhanced egg production in hens [12]. The polyphenol compounds can enhance the digestive enzymes, intestinal morphology, and microbiota, and as a result, increase intestinal digestion and absorption; these beneficial changes can ultimately improve the bird’s performance [27,28,29,30]. Contrary to our findings, chlorogenic acid in the diet of hens did not alter the performance significantly as compared to the control diet. In a previous study, flaxseed with antioxidants enhance the egg weight, which was in accordance with our study [31]. Dietary treatments having flaxseed with lutein did not affect feed intake and egg weight [32]. It is suggested that adding natural antioxidants such as VE, CA, PF, and lutein could enhance the egg weight and HDEP in laying hens.

In the egg, the albumen height and Haugh unit represent the quality of the albumen. Ovomucin is vital in defining the height of the thick albumen, and is responsible for the thick gel characteristics of liquid albumen [33]. The supplementation of polyphenols improved the albumen quality of laying hens [12]. The previous study [34] reported no change in the albumen quality when polyphenols were added to the quail’s diet. A previous antioxidant study reported better egg quality when chlorogenic acid and polyphenols were added to the hen’s diet [15]. In our study, all antioxidants maintained the egg quality during 42 days of storage, except CA, in which egg quality declined after 28 days of storage. The enhanced albumen quality during storage in the VE, PF, and L group might be due to the reduction in oxidative stress, as shown in the egg MDA, and an increase in the antioxidant enzymes, which could enhance the morphology of magnum tissues. The physical factors that affect the egg white quality are magnum fold height and magnum epithelium height [4,35]. The cilia height and movement are other factors that facilitate the egg white quality. During egg white formation, the quality might be declined by inhibition of magnum motility via cilia [36]. This statement was approved in our study, where a higher fold height, epithelium height, and cilia height were observed in the VE, PF, and L groups. This enhancement can be due to the antioxidant properties of the supplemented antioxidants. Our findings indicate that the polyphenols, lutein, and vitamin E can maintain the egg quality during storage and enhance the microstructures in the magnum of hens.

Despite the enrichment of egg yolk or chicken meat with n-3 FA and DHA, the n-3-enriched products are susceptible to oxidation [37]. It is also accepted that unsaturated fatty acids in the egg yolk are prone to lipid oxidation due to the longer chain length during the extended storage period and high ambient temperature [38]. Feeding flaxseed or fish oil in hen diets has been found to increase the thiobarbituric acid reactive substances (TBARS) and peroxide values in yolk [39]. To retard the lipid peroxidation in poultry products, the diet of the chickens must be supplemented with natural antioxidant compounds [15]. Lipid oxidation is the primary mechanism of the decline in egg yolk lipids [40]. The biomarker of lipid oxidation is malondialdehyde (MDA). The oxidation of lipids can increase the production and buildup of free radicals or reactive oxygen species (ROS) directly or by reducing the cell’s ability to eradicate ROS, thereby causing oxidative stress in cells [41]. The SOD, GSH-Px, and HO-1 are chief antioxidant enzymes that protect against ROS and each enzyme plays an integral role in redox balance modulation [42]. In our study, the control group increases MDA and decreases the antioxidant enzymes, such as GSH-Px, SOD, and T-AOC, in the yolk during the storage period. In agreement with our study, the yolk MDA was increased in the control group during storage [43]. It suggests that supplementation with antioxidants such as VE, PF, and L in the flaxseed diets reduces the yolk MDA contents and maintains the antioxidant defense enzymes GSH-Px, SOD, and T-AOC.

The egg yolk DHA profile suggests that during 42 days of storage, VE, PF, and CA maintained the DHA content, while lutein decreased the DHA content after 28 d of storage. The VE, CA, PF, and lutein groups did not decrease the ALA content during 42 days of storage. The total n-3 was maintained during 42 days of storage by VE, CA, and PF, except for lutein, which decreased the total n-3 after 35 days of storage. Contrary to our finding, the storage reduced the egg total n-3 and total n-6 fatty acids during the 20 days of storage, although VE was added to the diet [9]. A 29% reduction was observed in the total n-3 fatty acid content of FO eggs at Day 60 of storage when compared with Day 0 of storage [44]. The results suggest that supplementing VE, PF, and CA can maintain the egg yolk total n-3, ALA, and DHA during storage. The suggested intake of DHA by the European Food Safety Authority (EFSA) is 100 mg/d for young children, 200 mg/d for pregnancy and lactation, and 150–200 mg/d for adults (EFSA Panel on Dietetic Products and Allergies [45]). In this study, storage of eggs maintained 70–74 mg/egg DHA in the yolk. The intake of two n-3 eggs provides enough DHA contents for human consumption.

The heme-oxygenase (HO-1), a microsomal enzyme induced during oxidative stress, is responsible for the conversion of heme to biliverdin, carbon monoxide, and iron in the blood and shell gland [46]. HO-1 also exerts a protective effect against oxidative stress in various cells [47,48]. VE, PF, and L increase the expression of HO-1, SOD, and GSH-Px in the magnum and liver, which could prevent lipid oxidation and maintain the antioxidant enzymes during the egg-storage period. The enhancement of the magnum’s health might be due to these antioxidant defense enzymes. These antioxidant defense enzymes in the mRNA expression of the magnum and liver of hens suggest that these enzymes can perform a defensive part against oxidative damage caused by ROS [49].

Nrf2 is a key transcriptional factor that activates the antioxidant-reactive element (ARE), in turn regulating the expression of antioxidant phase II detoxifying enzymes [50]. Under normal physiological conditions, Nrf2 is bound to Keap1 in the cytoplasm; however, when the cellular redox balance is disrupted, Nrf2 is released from Keap1 and rapidly translocate to the nucleus to initiate transcription of antioxidant genes [51,52]. In the present study, the Nrf-2 gene was upregulated in the VE, PF, and L groups in the magnum and hepatic tissues. In accordance with our study, Wang et al. [12] also reported upregulation of Nrf2, HO-1, and GSH-Px. Previous studies reported that green tea polyphenols are a potent Nrf2 activator [53]. Moreover, the polyphenols can improve cellular antioxidant capacity by upregulating the production of Nrf-2-mediated phase II detoxification enzymes [52,54].

MAPKs such as P38 can certainly regulate Nrf2 translocation and Nrf2-targeted genes [19,42]. In this study, the VE, PF, and L groups upregulated the P38MAPK phosphorylation, which was decreased in the control group. Antioxidants such as VE, PF, and L increased the phosphorylation of Nrf-2 via the P38MAPK pathway, increased its nuclear translocation, and the antioxidant enzymes were enhanced, which could prevent ROS in these groups, as seen in the eggs during storage.

## 5. Conclusions

We conclude that supplemental VE, CA, PF, and L could increase the egg weight and egg production of laying hens fed with a flaxseed diet. VE, PF, and L could maintain the albumen height and Haugh unit and antioxidant defense enzymes during 42 days of storage. The antioxidants VE, PF, CA, and lutein maintained more than 300 mg/egg n-3 FA in the yolk during storage and are suitable for health-conscious consumers. PF and L are better to prevent egg quality deterioration and lipid oxidation, while PF and CA prevent FA loss during storage; these antioxidants can replace vitamin E in hens’ diets. VE, PF, and L increased the magnum fold height, epithelium height, and cilia height as compared to CA. We found a mechanism whereby the VE, PF, and L groups activated the Nrf-2 pathway through phosphorylation of P38MAPK, and enhanced the phase-2 antioxidant defense enzyme, namely, SOD, GSH-Px, and HO-1. Therefore, the use of PF and lutein is recommended in the flaxseed diet to prevent egg quality deterioration, lipid oxidation, and FA loss during storage.

## Figures and Tables

**Figure 1 foods-11-03158-f001:**
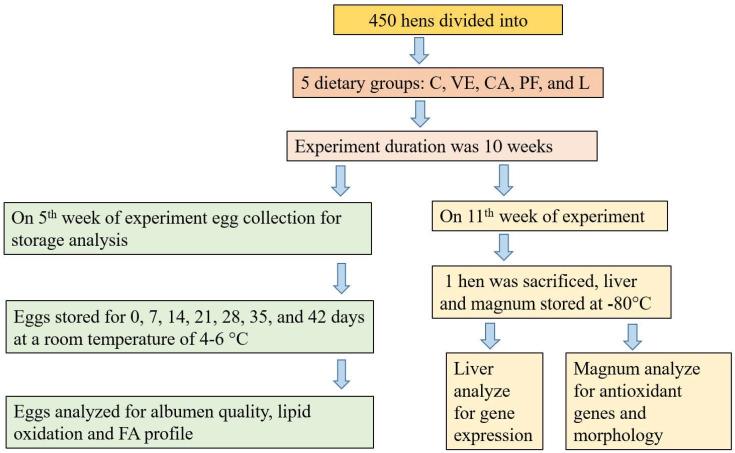
Graphical representation of the experimental scheme. C = control; VE 0.02% = vitamin E; CA 0.24% = chlorogenic acid; PF 0.05% = polyphenol; L 0.03% = lutein; FA profile = fatty acid profile.

**Figure 2 foods-11-03158-f002:**
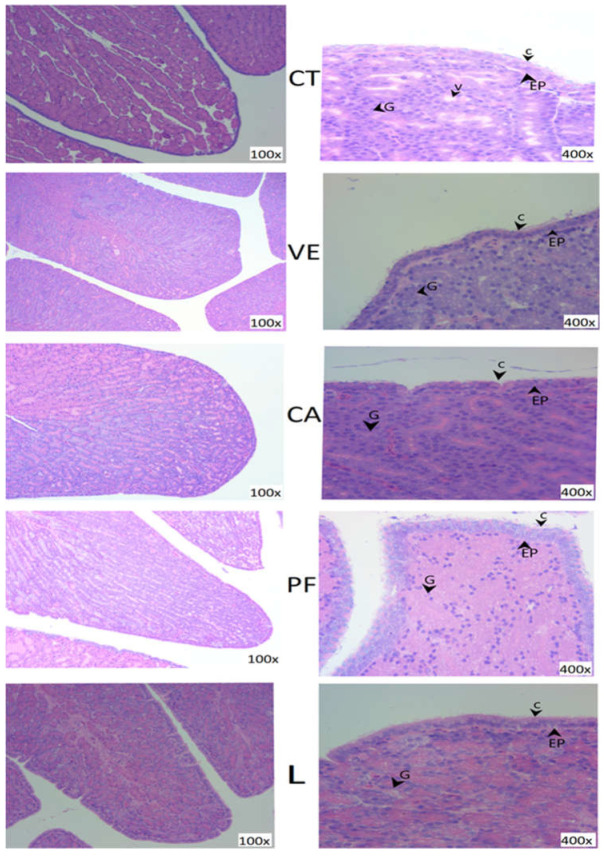
Hematoxylin-and-eosin-stained images of the cross section (100× magnification) or longitudinal section (400× magnification) of the magnum in laying hens (36 wk) after a 10-wk treatment period. EP = epithelium line of the mucosa; C = cilia; G = tubular glands; V = vacuole; control = CT; VE 0.02% = vitamin E; CA 0.24% = chlorogenic acid; PF 0.05% = polyphenol; and L 0.03% = lutein.

**Table 1 foods-11-03158-t001:** Ingredients composition.

Ingredients %	Composition, %	Nutrients	Content
Wheat	70	AME (kcal/kg)	2740
Soybean meal	7.9	Protein, %	15.99
Flaxseed	10	Lysine, %	0.79
Limestone	9.4	Methionine, %	0.38
Calcium hydro-phosphate	0.8	M + C, %	0.63
Salt	0.35	Ca, %	3.91
Choline chloride	0.1	NPP, %	0.26
Minerals ^1^	0.2		
DL-Met	0.13		
L-lysine	0.65		
Vitamin ^2^	0.02		
Phytase	0.02		
Compound enzyme ^3^	0.02		
Pigment compound	0.01		
Zeolite powder	0.4		

^1^ The trace mineral premix provided the following per kg of diets: Cu (CuSO_4_•5H_2_O), 8.00 mg; Zn (ZnSO_4_), 75.00 mg; Fe (FeSO_4_•H_2_O), 80.00 mg; Mn (MnSO_4_•H_2_O), 60.00 mg; Se (Na_2_SeO_3_), 0.30 mg; I (C(IO_3_)_2_), 0.35 mg. ^2^ The vitamin premix provided the following per kg of diets, vitamin A (trans-retinyl acetate) 9000 IU, vitamin D_3_ 2500 IU, vitamin E (DL-α-tocopherol) 10 IU, vitamin K_3_ 2.65 mg, vitamin B_1_ 2.00 mg, vitamin B_2_ 6.00 mg, vitamin B_6_ 6.00 mg, vitamin B12 0.03 mg, biotin 0.03 mg, folic acid 1.25 mg, pantothenic acid 12.00 mg, and nicotinic acid 20.00 mg. ^3^ Compound enzyme contains neutral protease 10,000, xylanase 35,000, β-mannanase 1500, β-glucanase 2000, cellulose 500, and amylase 100, pectinase 10,000 (U g^−^^1^).

**Table 2 foods-11-03158-t002:** Sequences of primer pairs of mRNA.

Genes	FORWARD	REVERSE
*Nrf2*	TGTGTGTGATTCAACCCGACT	TTAATGGAAGCCGCACCACT
*SOD1*	TTGTCTGATGGAGATCATGGCTTC	TGCTTGCCTTCAGGATTAAAGTGAG
*P38MAPK*	TGTGTTCACCCCTGCCAAGT	GCCCCCGAAGAATCTGGTAT
*HO-1*	TTGGCAAGAAGCATCCAGA	TCCATCTCAAGGGCATTCA
*GSH-Px*	TCACCATGTTCGAGAAGTGC	ATGTACTGCGGGTTGGTCAT
*CAT*	GTTGGCGGTAGGAGTCTGGTCT	GTGGTCAAGGCATCTGGCTTCTG
*beta-actin*	TCAGGGTGTGATGGTTGGTATG	TGTTCAATGGGGTACTTCAGGG

Nrf2 = nuclear factor erythroid-2 related factor 2; SOD = superoxide dismutase; P38 MAPK = p38 mitogen activated protein kinases; HO-1 = heme oxygenase-1; GSH-Px = glutathione peroxidase 1; CAT = catalase.

**Table 3 foods-11-03158-t003:** The effect of different antioxidants on laying hens’ performances.

Groups	Weeks	Control	VE	CA	PF	Lutein	SEM	*p*-Value ^1^
**Egg** **weight (g)**	**2 wks**	53.28 ^b^	56.33 ^a^	55.79 ^a^	56.11 ^a^	56.04 ^a^	0.285	0.001
	**4 wks**	53.30 ^b^	56.20 ^a^	56.38 ^a^	55.99 ^a^	55.38 ^a^	0.294	0.001
	**6 wks**	53.29 ^b^	56.29 ^a^	56.30 ^a^	55.83 ^a^	55.92 ^a^	0.281	<0.001
	**8 wks**	53.45 ^b^	56.14 ^a^	56.15 ^a^	55.30 ^a^	56.16 ^a^	0.271	0.001
	**10 wks**	54.22 ^b^	56.94 ^a^	56.48 ^a^	56.06 ^a^	56.55 ^a^	0.284	0.012
	*p*-value ^2^	0.611	0.495	0.756	0.855	0.700		
	Linear ^3^	0.202	0.293	0.366	0.666	0.330		
	Quadratic ^4^	0.369	0.222	0.689	0.521	0.407		
**Hen** **day** **egg** **production (%)**	**2 wks**	77.83 ^b^	82.79 ^a^	80.96 ^aC^	76.21 ^b^	83.60 ^aB^	0.500	<0.001
	**4 wks**	77.89 ^b^	86.32 ^a^	84.44 ^aB^	76.45 ^b^	86.34 ^aAB^	0.673	<0.001
	**6 wks**	77.88 ^b^	87.12 ^a^	89.10 ^aA^	76.40 ^b^	86.24 ^bAB^	0.753	<0.001
	**8 wks**	78.03 ^b^	85.79 ^a^	88.85 ^aA^	76.40 ^b^	90.35 ^aA^	0.770	<0.001
	**10 wks**	78.59 ^b^	85.23 ^a^	88.68 ^aA^	76.56 ^b^	91.11 ^aA^	1.260	<0.001
	*p*-value ^2^	0.999	0.654	<0.001	1.000	0.016		
	Linear ^3^	0.808	0.518	<0.001	0.870	0.001		
	Quadratic ^4^	0.886	0.199	0.011	0.977	0.969		
**Feed Intake (g)**	**2 wks**	84.48	84.82	85.08	84.14	84.65 ^A^	0.157	0.488
	**4 wks**	84.75 ^a^	85.15 ^a^	85.42 ^a^	83.60 ^b^	82.77 ^bB^	0.228	0.025
	**6 wks**	84.73	84.37	84.77	84.76	85.50 ^A^	0.146	0.645
	**8 wks**	84.82 ^b^	84.87 ^b^	84.95 ^b^	84.87 ^b^	85.45 ^aA^	0.090	0.005
	**10 wks**	85.59	84.85	85.18	85.00	85.73 ^A^	0.120	0.097
	*p*-value ^2^	0.149	0.062	0.543	0.195	0.001		
	Linear ^3^	0.027	0.696	0.767	0.051	0.004		
	Quadratic ^4^	0.346	0.387	0.558	0.867	0.403		
**Feed** **Conversion** **Ratio (%)**	**2 wks**	1.88	1.78	1.85	1.81	1.81	0.010	0.312
	**4 wks**	1.89	1.81	1.87	1.81	1.75	0.012	0.125
	**6 wks**	1.87	1.77	1.86	1.84	1.80	0.013	0.274
	**8 wks**	1.88	1.79	1.88	1.84	1.81	0.010	0.448
	**10 wks**	1.87	1.80	1.81	1.82	1.79	0.630	0.627
	*p*-value ^2^	0.987	0.933	0.748	0.833	0.499		
	Linear ^3^	0.725	0.733	0.557	0.512	0.877		
	Quadratic ^4^	0.892	0.890	0.304	0.464	0.623		

VE 0.02% = vitamin E; CA 0.24% = chlorogenic acid; PF 0.05% = polyphenol; L 0.03% = lutein; SEM = standard error of mean, *n* = 6. The small letters represent significant differences among treatments at each week’s time (rows); a post-hoc Duncan test was used for means comparison (*p*-values ^1^). The capitals letters represent significant differences among means for individual treatment from 2nd to 10th weeks period (columns); a post-hoc Duncan test (*p*-values ^2^), linear (*p*-values ^3^), and quadratic (*p*-values ^4^) polynomial contrasts were applied for comparisons. Significance was set at *p* < 0.05 for all tests.

**Table 4 foods-11-03158-t004:** The effect of different antioxidants on the egg quality during storage.

Groups	Storage	Control	VE	CA	PF	Lutein	SEM	*p*-Value ^1^
**Albumen** **height (mm)**	**0 d**	6.24 ^bA^	7.37 ^a^	7.42 ^aA^	7.47 ^a^	7.45 ^a^	0.14	0.012
	**7 d**	6.20 ^bA^	7.36 ^a^	7.33 ^aA^	7.28 ^a^	7.28 ^a^	0.15	0.016
	**14 d**	6.06 ^bA^	7.31 ^a^	7.39 ^aA^	7.41 ^a^	7.47 ^a^	0.11	<0.001
	**21 d**	5.55 ^bA^	7.22 ^a^	7.17 ^aA^	7.14 ^a^	7.24 ^a^	0.16	<0.001
	**28 d**	5.46 ^bA^	7.10 ^a^	6.12 ^bB^	7.01 ^a^	7.07 ^a^	0.15	<0.001
	**35 d**	4.17 ^cB^	6.88 ^a^	5.83 ^bBC^	6.92 ^a^	6.81 ^a^	0.21	<0.001
	**42 d**	4.00 ^cB^	6.71 ^a^	5.23 ^bC^	6.76 ^a^	6.65 ^a^	0.21	<0.001
	*p*-value ^2^	<0.001	0.153	<0.001	0.535	0.083		
	Linear ^3^	<0.001	0.005	<0.001	0.037	0.003		
	Quadratic ^4^	0.005	0.349	0.003	0.792	0.297		
**Haugh** **unit (AA)**	**0 d**	84.65 ^aA^	89.05 ^aA^	91.24 ^aA^	88.70 ^aA^	89.40 ^aA^	0.57	0.001
	**7 d**	83.73 ^bA^	89.23 ^aA^	90.32 ^aA^	89.02 ^aA^	88.99 ^aA^	0.58	<0.001
	**14 d**	81.62 ^bAB^	89.84 ^aA^	90.36 ^aA^	89.75 ^aA^	88.59 ^aA^	0.77	<0.001
	**21 d**	76.07 ^bB^	88.23 ^aA^	88.51 ^aA^	88.45 ^aA^	89.09 ^aA^	1.03	<0.001
	**28 d**	68.26 ^cC^	87.27 ^aA^	80.14 ^bB^	87.09 ^aAB^	87.30 ^aA^	1.56	<0.001
	**35 d**	61.25 ^cD^	83.61 ^aB^	75.31 ^bBC^	83.93 ^aBC^	84.80 ^aA^	1.80	<0.001
	**42 d**	57.92 ^cD^	82.27 ^aB^	73.81 ^bC^	81.59 ^aC^	80.13 ^aB^	1.77	<0.001
	*p*-value ^2^	<0.001	<0.001	<0.001	<0.001	<0.001		
	Linear ^3^	<0.001	<0.001	<0.001	<0.001	<0.001		
	Quadratic ^4^	0.008	0.004	0.007	0.002	0.002		

VE 0.02% = vitamin E; CA 0.24% = chlorogenic acid; PF 0.05% = polyphenol; L 0.03% = lutein; SEM = standard error of mean, *n* = 6. The small letters represent significant differences among treatments at one storage time (rows); a post-hoc Duncan test was used for means comparison (*p*-values ^1^). The capitals letters represent significant differences among means for individual treatment from 0–42 d of storage (columns); a post-hoc Duncan test (*p*-values ^2^), linear (*p*-values ^3^), and quadratic (*p*-values ^4^) polynomial contrasts were applied for comparisons. Significance was set at *p* < 0.05 for all tests.

**Table 5 foods-11-03158-t005:** The effect of different antioxidants on the egg oxidation parameters during storage.

Groups	Storage	Control	VE	CA	PF	Lutein	SEM	*p*-Value ^1^
**MDA (mg/protein)**	**0 d**	6.37 ^aD^	3.21 ^cB^	3.30 ^cB^	3.65 ^b^	3.25 ^cB^	0.227	<0.001
	**7 d**	6.39 ^aD^	3.22 ^cB^	3.30 ^cB^	3.65 ^b^	3.25 ^cB^	0.228	<0.001
	**14 d**	6.51 ^aD^	3.22 ^cB^	3.30 ^cB^	3.65 ^b^	3.25 ^cB^	0.236	<0.001
	**21 d**	7.42 ^aC^	3.23 ^cB^	3.31 ^cB^	3.65 ^b^	3.25 ^cB^	0.304	<0.001
	**28 d**	7.72 ^aC^	3.26 ^cB^	3.31 ^cB^	3.67 ^b^	3.26 ^cB^	0.325	<0.001
	**35 d**	8.44 ^aB^	3.26 ^cB^	3.33 ^bcB^	3.69 ^b^	3.28 ^cB^	0.379	<0.001
	**42 d**	8.94 ^aA^	3.53 ^bA^	3.49 ^bA^	3.74 ^b^	3.44 ^bA^	0.403	<0.001
	*p*-value ^2^	<0.001	0.014	0.034	0.575	<0.001		
	Linear ^3^	<0.001	0.003	0.009	0.078	<0.001		
	Quadratic ^4^	<0.001	0.028	0.032	0.908	0.002		
**SOD (mg/protein)**	**0 d**	109.57 ^bA^	123.37 ^a^	123.19 ^aA^	123.52 ^aA^	123.46 ^a^	1.035	<0.001
	**7 d**	109.60 ^bA^	123.28 ^a^	123.19 ^aA^	123.51 ^aA^	123.46 ^a^	1.031	<0.001
	**14 d**	108.57 ^bAB^	123.34 ^a^	123.20 ^aA^	123.52 ^aA^	123.46 ^a^	1.108	<0.001
	**21 d**	108.24 ^bAB^	123.38 ^a^	123.20 ^aA^	123.52 ^aA^	123.46 ^a^	1.133	<0.001
	**28 d**	107.57 ^bABC^	123.47 ^a^	123.21 ^aA^	123.51 ^aA^	123.46 ^a^	1.189	<0.001
	**35 d**	106.74 ^bC^	122.87 ^a^	123.09 ^aA^	123.25 ^aA^	123.27 ^a^	1.226	<0.001
	**42 d**	105.75 ^bC^	121.69 ^a^	122.03 ^aB^	122.18 ^aB^	122.15 ^a^	1.222	<0.001
	*p*-value ^2^	<0.001	0.050	0.008	0.008	0.062		
	Linear ^3^	<0.001	0.013	0.005	0.003	0.017		
	Quadratic ^4^	0.281	0.028	0.009	0.009	0.035		
**T-AOC (mg/protein)**	**0 d**	6.53 ^bA^	8.14 ^aA^	8.17 ^aA^	8.18 ^a^	8.24 ^aA^	0.128	<0.001
	**7 d**	6.52 ^bA^	8.14 ^aA^	8.16 ^aA^	8.17 ^a^	8.24 ^aA^	0.124	<0.001
	**14 d**	6.51 ^cA^	8.11 ^bA^	8.13 ^abAB^	8.17 ^ab^	8.24 ^aA^	0.123	<0.001
	**21 d**	6.24 ^bAB^	8.11 ^aA^	8.11 ^aAB^	8.14 ^a^	8.21 ^aA^	0.149	<0.001
	**28 d**	5.91 ^bBC^	8.10 ^aA^	8.11 ^aAB^	8.12 ^a^	8.20 ^aA^	0.166	<0.001
	**35 d**	5.70 ^bC^	8.03 ^aA^	7.93 ^aBC^	8.08 ^a^	8.11 ^aA^	0.176	<0.001
	**42 d**	4.97 ^bD^	7.76 ^aB^	7.76 ^aC^	7.99 ^a^	7.97 ^aB^	0.219	<0.001
	*p*-value ^2^	<0.001	<0.001	<0.001	0.068	<0.001		
	Linear ^3^	<0.001	<0.001	<0.001	0.002	<0.001		
	Quadratic ^4^	<0.001	0.002	0.004	0.187	0.003		
**GSH-Px (mg/protein)**	**0 d**	767.51 ^bA^	992.42 ^aA^	991.61 ^aA^	988.95 ^aA^	990.99 ^aA^	16.63	<0.001
	**7 d**	767.25 ^bA^	992.35 ^aA^	991.52 ^aA^	988.60 ^aA^	990.62 ^aA^	16.63	<0.001
	**14 d**	766.38 ^bA^	991.37 ^aAB^	991.37 ^aA^	988.54 ^aA^	990.57 ^aA^	16.67	<0.001
	**21 d**	763.41 ^bA^	990.20 ^aAB^	991.34 ^aA^	988.50 ^aA^	990.42 ^aA^	16.87	<0.001
	**28 d**	757.65 ^bA^	988.54 ^aAB^	991.20 ^aA^	987.87 ^aA^	990.37 ^aA^	17.26	<0.001
	**35 d**	687.58 ^bB^	982.42 ^aB^	980.37 ^aB^	979.70 ^aB^	981.40 ^aB^	21.80	<0.001
	**42 d**	684.25 ^bB^	970.20 ^aC^	971.87 ^aC^	972.70 ^aB^	977.73 ^aB^	21.54	<0.001
	*p*-value ^2^	<0.001	<0.001	<0.001	<0.001	0.005		
	Linear ^3^	<0.001	<0.001	<0.001	<0.001	<0.001		
	Quadratic ^4^	<0.001	<0.001	<0.001	0.001	0.023		

VE 0.02% = vitamin E; CA 0.24% = chlorogenic acid; PF 0.05% = polyphenol; L 0.03% = lutein; SEM = standard error of mean, *n* = 6. The small letters represent significant differences among treatments at one storage time (rows); a post-hoc Duncan test was used for means comparison (*p*-values ^1^). The capitals letters represent significant differences among means for individual treatment from 0–42 d of storage (columns); a post-hoc Duncan test (*p*-values ^2^), linear (*p*-values ^3^), and quadratic (*p*-values ^4^) polynomial contrasts were applied for comparisons. Significance was set at *p* < 0.05 for all tests.

**Table 6 foods-11-03158-t006:** The effect of different antioxidants on the DHA, ALA, Tn-3, and Tn-6 of egg yolk during storage, expressed as mg/egg.

FA	Storage	Control	VE	CA	PF	L	SEM	*p*-Value ^1^
**DHA**	0 d	65.10 ^bA^	74.81 ^a^	73.78 ^a^	74.15 ^a^	74.28 ^aA^	0.71	<0.001
	7 d	65.10 ^bA^	74.70 ^a^	73.77 ^a^	74.11 ^a^	74.09 ^aA^	0.71	<0.001
	14 d	64.70 ^bA^	74.53 ^a^	73.43 ^a^	74.09 ^a^	74.18 ^aA^	0.73	<0.001
	21 d	64.19 ^bA^	74.33 ^a^	73.25 ^a^	74.06 ^a^	73.66 ^aA^	0.76	<0.001
	28 d	60.83 ^bB^	74.16 ^a^	73.05 ^a^	73.53 ^a^	72.68 ^aAB^	0.96	<0.001
	35 d	57.87 ^cC^	73.96 ^a^	72.92 ^a^	73.01 ^a^	70.70 ^bBC^	1.15	<0.001
	42 d	52.69 ^cD^	73.38 ^a^	72.86 ^a^	73.08 ^a^	70.07 ^bC^	1.53	<0.001
	*p*-value ^2^	<0.001	0.519	0.911	0.514	<0.001		
	Linear ^3^	<0.001	0.033	0.171	0.043	<0.001		
	Quadratic ^4^	<0.001	0.612	0.909	0.549	0.042		
**ALA**	0 d	324.39 ^bA^	332.41 ^a^	331.46 ^a^	333.12 ^a^	331.76 ^a^	0.75	<0.001
	7 d	324.25 ^bA^	332.32 ^a^	331.43 ^a^	333.14 ^a^	331.67 ^a^	0.74	<0.001
	14 d	324.07 ^bA^	332.23 ^a^	331.28 ^a^	333.07 ^a^	331.48 ^a^	0.75	<0.001
	21 d	322.52 ^bA^	332.29 ^a^	331.19 ^a^	332.99 ^a^	331.40 ^a^	0.85	<0.001
	28 d	319.32 ^bB^	331.71 ^a^	330.82 ^a^	332.64 ^a^	331.41 ^a^	1.02	<0.001
	35 d	315.75 ^bC^	331.04 ^a^	330.00 ^a^	331.47 ^a^	330.74 ^a^	1.20	<0.001
	42 d	312.55 ^bD^	329.74 ^a^	329.56 ^a^	331.00 ^a^	329.41 ^a^	1.36	<0.001
	*p*-value ^2^	<0.001	0.380	0.579	0.955	0.581		
	Linear ^3^	<0.001	0.032	0.050	0.284	0.075		
	Quadratic ^4^	<0.001	0.896	0.983	0.997	0.921		
**Total n-3**	0 d	426.41 ^bA^	436.49 ^a^	435.34 ^a^	437.55 ^a^	435.80 ^aA^	0.87	<0.001
	7 d	426.28 ^bA^	436.28 ^a^	435.15 ^a^	437.37 ^a^	435.70 ^aA^	0.88	<0.001
	14 d	426.09 ^bA^	436.25 ^a^	435.12 ^a^	437.15 ^a^	435.44 ^aA^	0.88	<0.001
	21 d	424.69 ^bAB^	436.06 ^a^	435.01 ^a^	437.10 ^a^	435.05 ^aA^	0.97	<0.001
	28 d	423.63 ^bB^	435.61 ^a^	434.67 ^a^	436.18 ^a^	434.55 ^aA^	0.96	<0.001
	35 d	419.24 ^bC^	435.34 ^a^	434.66 ^a^	435.70 ^a^	433.89 ^aA^	1.24	<0.001
	42 d	409.94 ^cD^	434.82 ^a^	433.50 ^ab^	434.65 ^a^	431.15 ^bB^	1.82	<0.001
	*p*-value ^2^	<0.001	0.934	0.823	0.845	0.005		
	Linear ^3^	<0.001	0.206	0.152	0.133	<0.001		
	Quadratic ^4^	<0.001	0.751	0.529	0.618	0.050		
**Total n-6**	0 d	557.63 ^aA^	549.73 ^c^	551.57 ^bcA^	553.95 ^bA^	551.61 ^bc^	0.64	<0.001
	7 d	557.59 ^aA^	549.58 ^c^	551.56 ^bcA^	553.75 ^bA^	551.57 ^bc^	0.63	<0.001
	14 d	557.41 ^aA^	549.39 ^c^	551.40 ^bcA^	553.55 ^bA^	551.24 ^bc^	0.64	<0.001
	21 d	556.57 ^aAB^	549.42 ^c^	551.37 ^bcA^	552.92 ^bA^	551.10 ^bc^	0.61	0.001
	28 d	555.21 ^aBC^	548.71 ^b^	550.89 ^bAB^	552.25 ^abAB^	550.57 ^b^	0.66	0.018
	35 d	554.53 ^aCD^	547.56 ^b^	547.93 ^bBC^	550.92 ^abB^	548.77 ^b^	0.79	0.018
	42 d	552.97 ^D^	546.47	546.95 ^C^	547.92 ^C^	547.74	0.82	0.073
	*p*-value ^2^	<0.001	0.468	0.018	<0.001	0.663		
	Linear ^3^	<0.001	0.037	0.001	<0.001	0.074		
	Quadratic ^4^	0.058	0.346	0.064	0.016	0.441		

VE 0.02% = vitamin E; CA 0.24% = chlorogenic acid; PF 0.05% = polyphenol; L 0.03% = lutein; SEM = standard error of mean, *n* = 6. The small letters represent significant differences among treatments at one storage time (rows); a post-hoc Duncan test was used for means comparison (*p*-values ^1^). The capitals letters represent significant differences among means for individual treatment from 0–42 d of storage (columns); a post-hoc Duncan test (*p*-values ^2^), linear (*p*-values ^3^), and quadratic (*p*-values ^4^) polynomial contrasts were applied for comparisons. Significance was set at *p* < 0.05 for all tests.

**Table 7 foods-11-03158-t007:** The effect of different antioxidants on the relative mRNA gene expression of laying hens.

Tissue	Genes	Control	VE	CA	PF	L	SEM	*p*-Value
**Magnum**	**HO-1**	0.53 ^c^	1.09 ^a^	0.88 ^b^	1.05 ^a^	1.09 ^a^	0.044	<0.001
	**SOD1**	0.47 ^c^	1.19 ^a^	0.89 ^b^	1.20 ^a^	1.14 ^a^	0.054	<0.001
	**GSH-Px**	0.44 ^c^	0.89 ^a^	0.65 ^b^	0.98 ^a^	0.94 ^a^	0.045	<0.001
	**CAT**	0.51 ^c^	1.03 ^a^	0.71 ^b^	1.07 ^a^	1.00 ^a^	0.046	<0.001
	**Nrf-2**	0.63 ^c^	1.21 ^a^	0.93 ^b^	1.18 ^a^	1.17 ^a^	0.049	<0.001
	**P38MAPK**	0.55 ^c^	1.24 ^a^	0.96 ^b^	1.28 ^a^	1.26 ^a^	0.058	<0.001
**Liver**	**HO-1**	0.46 ^c^	1.13 ^a^	0.87 ^b^	1.11 ^a^	1.15 ^a^	0.052	<0.001
	**SOD1**	0.61 ^c^	1.22 ^a^	0.92 ^b^	1.21 ^a^	1.21 ^a^	0.050	<0.001
	**GSH-Px**	0.41 ^c^	0.85 ^a^	0.62 ^b^	0.95 ^a^	0.91 ^a^	0.045	<0.001
	**CAT**	1.17	1.18	1.16	1.17	1.14	0.016	0.929
	**Nrf-2**	0.55 ^c^	1.15 ^a^	0.84 ^b^	1.13 ^a^	1.12 ^a^	0.051	<0.001
	**P38MAPK**	0.90 ^c^	1.41 ^a^	1.20 ^b^	1.48 ^a^	1.52 ^a^	0.052	<0.001

VE 0.02% = vitamin E; CA 0.24% = chlorogenic acid; PF 0.05% = polyphenol; L 0.03% = lutein; SEM = standard error of mean, *n* = 6. The small letters in each row represent significant differences among treatments; a post-hoc Duncan test was used for means comparison. Significance was set at *p* < 0.05.

**Table 8 foods-11-03158-t008:** The effect of different antioxidants on the magnum morphology of laying hens.

Measures (μm)	Control	VE	CA	PF	Lutein	SEM	*p*-Value
**Fold height**	1692.02 ^c^	3520.30 ^a^	2592.00 ^b^	3773.10 ^a^	3652.03 ^a^	159.86	<0.001
**EP height**	164.58 ^c^	263.45 ^a^	204.40 ^b^	269.81 ^a^	266.42 ^a^	8.10	<0.001
**Cilia height**	5.37 ^c^	11.57 ^a^	8.40 ^b^	12.16 ^a^	11.45 ^a^	0.52	<0.001

VE 0.02% = vitamin E; CA 0.24% = chlorogenic acid; PF 0.05% = polyphenol; L 0.03% = lutein; SEM = standard error of mean, *n* = 6. The small letters in each row represent significant differences among treatments; a post-hoc Duncan test was used for means comparison. Significance was set at *p* < 0.05.

## Data Availability

The data presented in this study are available on request from the corresponding author.

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
