# Peer review of "Phytogenic Antioxidants Prolong n-3 Fatty Acid-Enriched Eggs’ Shelf Life by Activating the Nrf-2 Pathway through Phosphorylation of MAPK"

_foods, 2022, doi:10.3390/foods11203158_

Round 1

Reviewer 1 Report

The manuscript presented for review presents an ever-important problem related to the low intake of omega-3 fatty acids among the public, the research shown in the paper attempts to enrich eggs with acids from this group. The problem with most of the available sources of fatty acids is that they are susceptible to oxidation, so trying to find suitable and effective antioxidants is justified.

I have some comments wchich are as follows:

- lines 21 -22 - abbreviations should be explained when used for the first time

- Materials and methods - I suggest that the authors additionally consider a graphical representation of the experimental scheme.

- Results - as flaxseed is a rich source of alpha-linoleic acid, a precursor of EPA and DHA, why no results concerning the content of this acid are presented? THis would be interesting to analyze also the content of ALA in particular groups, especially that its conversion rate to EPA and DHA may differ

- DIscussion - line 227 - Flaxseed simply has the highest content of ALA among plant sources of n-3 acids. I suggest rearrange the sencence

- Table 6 - the title should be changed, the data in the table only concerns DHA and total n-3

- conclusion part should be separated from the Discussion

Author Response

AUTHOR’S  QUERY SHEET

Author(s): Muhammad Suhaib Shahid et al.

Article title: The phytogenic antioxidants prolong n-3 fatty acid-enriched egg's shelf life by activating the Nrf-2 pathway through phos-phorylation of MAPK

Dear editor

We are thankful to you for your overall review of our manuscript. We have made the point to point to answers to your queries as follows

AQ1      - lines 21 -22 - abbreviations should be explained when used for the first time

Response:  Abbreviations were explained in lines 21-22 and some other lines which were used for the first time.

AQ2      - Materials and methods - I suggest that the authors additionally consider a graphical representation of the experimental scheme.

Response: The graphical representation of the experimental scheme is added in Figure 1.

AQ3      why no results concerning the content of the ALA are presented?

Response: The results concerning the content of the ALA are presented in table 6.

AQ4      DIscussion - line 227 - Flaxseed simply has the highest content of ALA among plant sources of n-3 acids. I suggest rearranging the sentence.

Response: The sentence has been rearranged.

AQ5      - Table 6 - the title should be changed, the data in the table only concerns DHA and total n-3

Response: The title of table 6 has been corrected accordingly.

AQ6      - conclusion part should be separated from the Discussion

Response: The conclusion part is separated from the Discussion.

Reviewer 2 Report

It is exciting research with practical application. However, although it was well designed and done, there is a lack of information on essential details in materials and methods. The discussion must also be improved, also the English writing of the article. 

Author Response

AUTHOR’S  QUERY SHEET

Author(s): Muhammad Suhaib Shahid et al.

Article title: The phytogenic antioxidants prolong n-3 fatty acid-enriched egg's shelf life by activating the Nrf-2 pathway through phos-phorylation of MAPK

Dear editor

We are thankful to you for your overall review of our manuscript. We have made the point to point to answers to your queries as follows

AQ1 It is exciting research with practical application. However, although it was well designed and done, there is a lack of information on essential details in materials and methods. The discussion must also be improved, also the English writing of the article. 

Response: Material and methods were corrected and lacking information as described. The discussion was improved and the whole document was checked for English writing and grammar.

AQ2      Commented [MEO1]: Prevention of lipid oxidation is not just essential because of consumers' acceptance. Also, there is the production of toxic compounds from the oxidation of omega 3 during transport, storage, and consumption (Nogueira, S.M. et al. 2019. LWT-Food Sci. Technol. 101:113.122. https://doi.org/10.1016/j.lwt.2018.11.044).  

Response: This valuable reference was added in the introduction and the reference was numbered 5 in the text and reference section.

AQ3      Commented [MEO2]: Has it been written as a question? Why?

Response: This mistake was corrected.

AQ4      Commented [DMEOC3]: What was the concentration of the Lutein product?

Response: The concentration of Lutein is added as per company recommendations.

AQ5      Commented [DMEOC4]: What was the concentration in IU or mg?

Response: The concentration of VE and its united is added.

AQ6      Commented [DMEOC5]: How were animals kept on the floor or in battery cages?

Response: The were animals kept in the battery cages. The arrangements of hens were explained in the bird husbandry section.

AQ7      Commented [DMEOC6]: Vitamin E (VE), Chlorogenic acid (CA). polyphenol (PF) and Lutein (L)

Response: information added accordingly.

AQ8:       Commented [DMEOC7]: Why were these amounts chosen?

Response: The level of antioxidants was selected according to the company's recommended levels.

AQ9:      Commented [DMEOC8]: This is not clear

Response: It was corrected

AQ10      Commented [DMEOC9]: Lutein.

Response:  The spelling of Lutein was corrected

AQ11    Commented [DMEOC10]: As which? Commented [DMEOC11]: microbiota

Response:  The sentence was rephrased.

AQ12    Commented [DMEOC11]: microbiota

Response: microflora replaced with microbiota

AQ13     Commented [DMEOC12]: According to your results which of the antioxidants you investigated, whether you would conclude were the ones with the best results?

Response: Therefore, the use of VE, PF, and Lutein is recommended in the flaxseed diet to prevent egg quality deterioration, lipid oxidation, and FA loss during storage.

AQ14       Commented [DMEOC13]: What was the composition of the vitamin mix?

Response: The composition of the vitamin mix and mineral mix was added in table 1.

AQ15       Commented [DMEOC14]: What were the other enzymes given besides phytase?

Response: The compound enzymes were added in table 1.
